# Freestanding Flexible Sensor Based on 3*ω* Technique for Anisotropic Thermal Conductivity Measurement of Potassium Dihydrogen Phosphate Crystal

**DOI:** 10.3390/s21237968

**Published:** 2021-11-29

**Authors:** Lin Qiu, Yuhao Ma, Yuxin Ouyang, Yanhui Feng, Xinxin Zhang

**Affiliations:** School of Energy and Environmental Engineering, University of Science and Technology Beijing, Beijing 100083, China; g20208226@xs.ustb.edu.cn (Y.M.); Ouyang64569@foxmail.com (Y.O.); yhfeng@me.ustb.edu.cn (Y.F.); xxzhang@ustb.edu.cn (X.Z.)

**Keywords:** freestanding flexible sensor, thermal conductivities, anisotropic

## Abstract

A new freestanding sensor-based 3ω technique is presented here, which remarkably expands the application of traditional 3ω technology to anisotropic materials. The freestanding flexible sensor was fabricated using the mature flexible printed circuit production technique, which is non-destructive to the samples and applicable to porous surfaces. The thermal conductivities of potassium dihydrogen phosphate (KDP) crystal along the (100), (010) and (001) crystallographic planes were measured based on this new sensor at room temperature. We found that the freestanding flexible sensor has considerable application value for thermal properties’ characterization for crystals with anisotropic thermophysical properties and other structures for which the traditional 3ω technique is not applicable.

## 1. Introduction

During the past three decades, large amounts of research have been devoted to the exploration of nonlinear optical crystal materials, and some excellent nonlinear optical crystals have been identified. Among them, potassium dihydrogen phosphate (KH${}_{2}$PO${}_{4}$ (KDP)) crystals are known as the most suitable material type for laser frequency doubling and optoelectronic switches in strong laser devices, and thus is widely used in nonlinear optical fields. With the development of high-power lasers, the heat dissipation issue becomes an important design consideration. Thermophysical properties, such as the thermal expansion coefficient and thermal conductivity, play an important role in the design of novel electronic devices and high-power laser systems [1,2]. However, it is difficult to accurately measure thermal properties because of the strong anisotropy of thermal conduction in KDP crystal. Hence, thermal conductivity measurement is important in the thermal design of laser crystal cooling systems.

Generally, thermal conductivity measurement methods for isotropic materials can be extended to measure anisotropic materials by varying the test configurations [3]. For example, the laser-flash method can be used to obtain both in-plane and cross-plane thermal diffusivities of anisotropic graphite, but the procedures require multiple samples, which is troublesome, and the samples often suffer damage when an over-high-power pulsed laser is used [4]. The photothermal method has also been used to measure anisotropic thermal conductivities, but the requirement for high spatial resolution prevents its wide applicability [5]. Although it is a suitable candidate for anisotropic thermal characterization, the conventional 3ω method based on harmonic voltage detection is destructive to samples because the metal sensor needs to be deposited on the samples by either lithography or vapor deposition, and cannot be applied to porous materials. Given these difficulties, suitable techniques to implement nondestructive anisotropic thermal conductivity measurement need to be developed.

Advances in the flexible printed circuit (FPC) production and fractal design [6] have created the possibility of fabricating new thin-film-based heaters/sensors to improve the 3ω technique [7]. Here, we fabricated a freestanding flexible sensor consisting of three four-pad patterns with different strip widths based on the FPC production technique. A piece of retaining tape is used to apply some pressure on the test structure to ensure an intimate contact between the sensor and the sample. According to our previous validation, when the appropriate loading pressure is used, the deviation in the obtained thermal conductivity produced by the thermal contact resistance between the sensor and the sample is small enough, so can be neglected [8]. Significantly, the sensor is not destroyed by the 3ω technique, which broadens its applicability to porous materials.

## 2. Freestanding Sensor Fabrication

Samples with typical dimensions of 10 × 3 × 3 cm were cut from a large KDP crystal along the crystal axes. The freestanding sensor used in this study is composed of two pieces of Kapton MT film encapsulating three nickel patterns in a configuration of a strip connected with four pads. First, three 150 nm thick nickel patterns with different widths (100, 40, and 10 µm) were deposited on a piece of 25 µm thick Kapton MT film using lithography and ultraviolet exposure processes. Then, four enameled wires were welded to the four pads for each pattern, which were used to connect to peripheral circuits. Finally, another piece of Kapton HN film was hot-pressed onto the patterns to complete the encapsulation. Our freestanding flexible sensor is thin enough, which can achieve flexible measurement and facilitate the heat waves penetrating through the film to detect the samples. A symmetrical test structure (sample–Kapton–metal strip–Kapton–sample) was used, which means the heat flux was half for each side. Thus, this reduced the effort to consider the bidirectional nature of heat spreading of the supported 3ω sensor [9]. The freestanding flexible sensor can measure block materials, porous thermal insulation materials, anisotropic materials with complex structures, and biological tissues such as skin [10] due to its nondestructiveness of the samples and applicability to porous surfaces, which expands the applications of 3ω technology. The schematic for the test structure and a photo of the freestanding flexible sensor are shown in Figure 1a,b, respectively.

The three patterns were deposited on the (001) crystalline plane of the KDP crystal sample, and the 40 µm sample was arranged vertical to the (010) crystalline plane (defined as *x* axis direction), and those that were 100 µm and 10 µm were arranged vertical to the (100) crystalline plane (defined as the *y*-axis direction). The thickness of the Kapton HN film is thin enough that the thermal wave from the sensor can easily penetrate the Kapton film and then proceed to the sample. The nickel patterns have a 1.6-times-larger thermal coefficient of resistance (TCR) than platinum in the temperature range of 0–250 ${}^{{}^{\circ}}$C, which increases sensitivity [11]. The freestanding flexible sensor works normally within 0 to 200 ${}^{{}^{\circ}}$C, which can meet the temperature requirements of the sample to be measured.

## 3. Experimental Methods

In brief, an alternating current (AC) with an angular frequency of 1ω coupled with a direct current offset component was applied to the three patterns during the test. The sensor served as both a heater and a thermometer without using any additional materials [12]. According to the Joule effect, a thermal wave with a frequency of 2ω was generated and propagate in the KDP crystal, resulting in a reduced temperature rise of the patterns and, thus, an electrical resistance increase of the patterns with a frequency of 2ω. Together with the AC current at a 1ω frequency, the voltage of the patterns had a 3ω angular frequency, which is related to temperature fluctuations. Figure 1c demonstrates our home-made electrical circuit for the 3ω technique. A 7265 phase-locked amplifier was used to record the third and first harmonic voltages of the patterns. The temperature oscillation of the patterns can be calculated according to [13]:(1)$$\Delta T=\frac{2U_{3\omega }}{\alpha_{\mathrm{cR}}U_{1\omega }}$$
where $\alpha_{\mathrm{cR}}$ is the TCR of the nickel strip, which is 0.0064 ($K^{-1}$) according to our experiments and suggests the change in resistance *R* with temperature *T* [14]; $U_{1\omega }$ is the amplitude of the voltage applied across the metal strip; and $U_{3\omega }$ is the amplitude of the measured voltage at 3ω. All voltages are described in root mean square (rms).

When the heating frequency was low, the thermal wave penetration depth in the sample was relatively large. Hence, the influence of the heat capacity of the patterns with a thickness of 150 nm could be ignored, and their temperature increase depended on the thermophysical properties of the sample. According to Cahill’s hypothesis [13], if the heat penetration depth is larger than five times the half-width of the pattern, the simplified formula is valid, which is called the slope 3ω method. Based on Borca’s analytical solution for the average temperature rise of the sensor [15], we only retain the first term of the infinite series from the expansion of Borca’s equation, which provides:(2)$$\Delta T^{*}=\frac{P_{l}}{\pi \sqrt{\lambda_{y}\lambda_{z}}}\left[0.5\ln\frac{\lambda_{y}}{Cb^{2}}-0.5\ln\left(\omega \right)+\eta -\frac{i\pi }{4}\right]$$
(3)$$\Delta T^{*}=\frac{P_{l}}{\pi \sqrt{\lambda_{x}\lambda_{z}}}\left[0.5\ln\frac{\lambda_{x}}{Cb^{2}}-0.5\ln\left(\omega \right)+\eta -\frac{i\pi }{4}\right]$$
where $P_{l}$ is the amplitude of the AC heating power per unit length of patterns; *i* is the imaginary component; ω is the heating frequency; *C* is the volume of heat capacity; *b* is the half-width of the patterns; η is a constant; and $\lambda_{x}$, $\lambda_{y}$, and $\lambda_{z}$ are the thermal conductivity in the x, y, and z directions, respectively. Note that the above equations have exactly the same form as Mishra’s solution [16] when $k_{xy}$ = 0. Mishra’s solution is more complex and able to obtain all thermal anisotropic conductivity tensor elements, but ours is relatively simple and was thus used to extract 3 anisotropic thermal conductivity tensor elements.

## 4. Results and Discussion

For the 100 µm strip width pattern, because the strip width is greater than the thermal penetration depth above a certain frequency, the thermal transport is still regarded as 1D along radial direction according to the study by Cahill. The heat is basically transferred along the direction vertically to (001) plane. Low-frequency experimental 3ω data were used for directly extracting the z-direction thermal conductivity for KDP crystal. In the low-frequency range, the results from narrow sensors were used to characterize the remaining in-plane thermophysical properties in the 0.010–0.1 Hz range. For the 10 and 40 µm strip width patterns, because the penetration depth of the thermal waves met the requirement of being at least five times larger than the heater half width, it can be regarded as a linear heat source. Heat would spread along both the in-plane and out-of-plane directions. We calculated the thermal conductivity in the radial direction based on the 100 µm strip width detector, and then calculated the thermal conductivity in other directions based on this radial thermal conductivity and the other two width detectors [17].

We used the surface impedance to compare the experimental results. The surface impedance is the ratio of the temperature rise to the heat flux (defined by *Z* = ΔT/q) [18]. The calculated surface impedances for three measurements are shown in Figure 2a. Theoretically, three slopes exist for the surface impedance rise curve for multilayer test structures. The small slope of the surface impedance versus the natural logarithm of frequency at low frequencies is the result of the high thermal conductivity of KDP crystal (the 10 µm strip width sensor has the smallest slope); for medium frequencies, the slope is steeper because of the low thermal conductivity of Kapton film. The slope again becomes small for high frequencies as the thermal wave remains inside the nickel strip. The slope of thermal impedance versus the logarithm of frequency in the low-frequency range (below 0.1 Hz) shows a decreasing trend as the half width of the nickel strip decreases, indicating that heat is transferred in-plane as the width of the nickel strip decreases. In the 0.01–0.1 Hz low-frequency range, the thermal wave travels far away from the Kapton HN film; thus, it is regarded as a line heat source for the semi-infinite samples. The simplicity and accuracy of the low-frequency slope method were used to analyze the data. Figure 2b shows the evolution trends of temperature rise versus frequency for three patterns. For the 100 µm strip width pattern, based on linear fitting of the temperature rise versus frequency in the low-frequency-region, the thermal conductivity in the z direction ($\lambda_{z}$) is obtained. For the 40 µm strip width pattern, the average thermal conductivity $\sqrt{\lambda_{y}\lambda_{z}}$ was obtained by linear fitting of temperature rise versus frequency according to Equation (Equation 2). After substituting the thermal conductivity in the z direction into $\sqrt{\lambda_{y}\lambda_{z}}$, the thermal conductivity in the y direction $\lambda_{y}$ was obtained. Similarly, for the 10 µm strip width pattern, the average thermal conductivity $\sqrt{\lambda_{x}\lambda_{z}}$ was obtained by linear fitting the temperature rise versus frequency according to Equation (Equation 3). After substituting the thermal conductivity in the z direction into $\sqrt{\lambda_{x}\lambda_{z}}$, the thermal conductivity in the x direction $\lambda_{x}$ was obtained. The thermal conductivities of KDP crystal in three directions are summarized in Table 1. The measured values using the present technique agree well with reference values using the laser flash method reported by Wang et al. [19], which validates the reliability and accuracy of the present technique.

## 5. Conclusions

In summary, we developed a new freestanding flexible sensor integrated with three patterns that is able to simultaneously measure anisotropic thermal conductivities. The reliability and accuracy were proven by measuring the anisotropic thermal conductivities of KDP crystals in the x, y, and z directions. The progress achieved shows that this freestanding flexible sensor has considerable application value for thermal properties’ characterization for crystals with anisotropic thermophysical properties and other structures for which the traditional 3ω technique is not applicable.

## Figures and Tables

**Figure 1 sensors-21-07968-f001:**
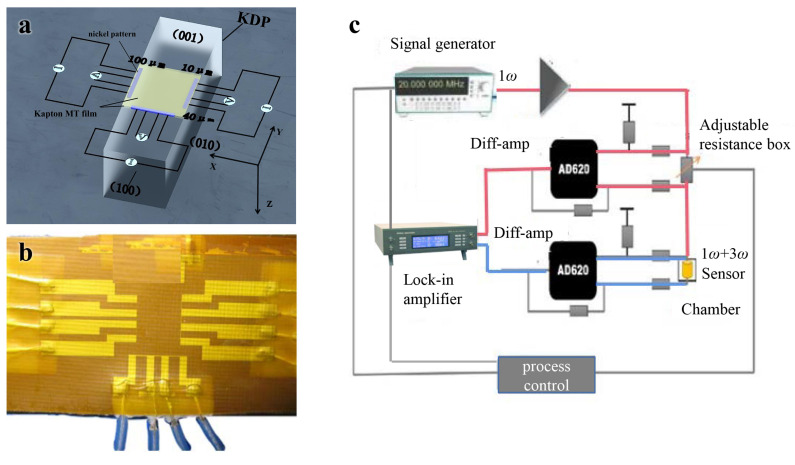
(**a**) Schematic diagram of three sensor on a KDP sample of different widths (100, 40, and 10 µm) that was 150 nm thick and 200 µm wide. (**b**) Photo of the freestanding flexible sensor. (**c**) The key components of the measurement system including signal generator, phase-locked amplifier, adjustable resistance, control circuit, and freestanding flexible sensor.

**Figure 2 sensors-21-07968-f002:**
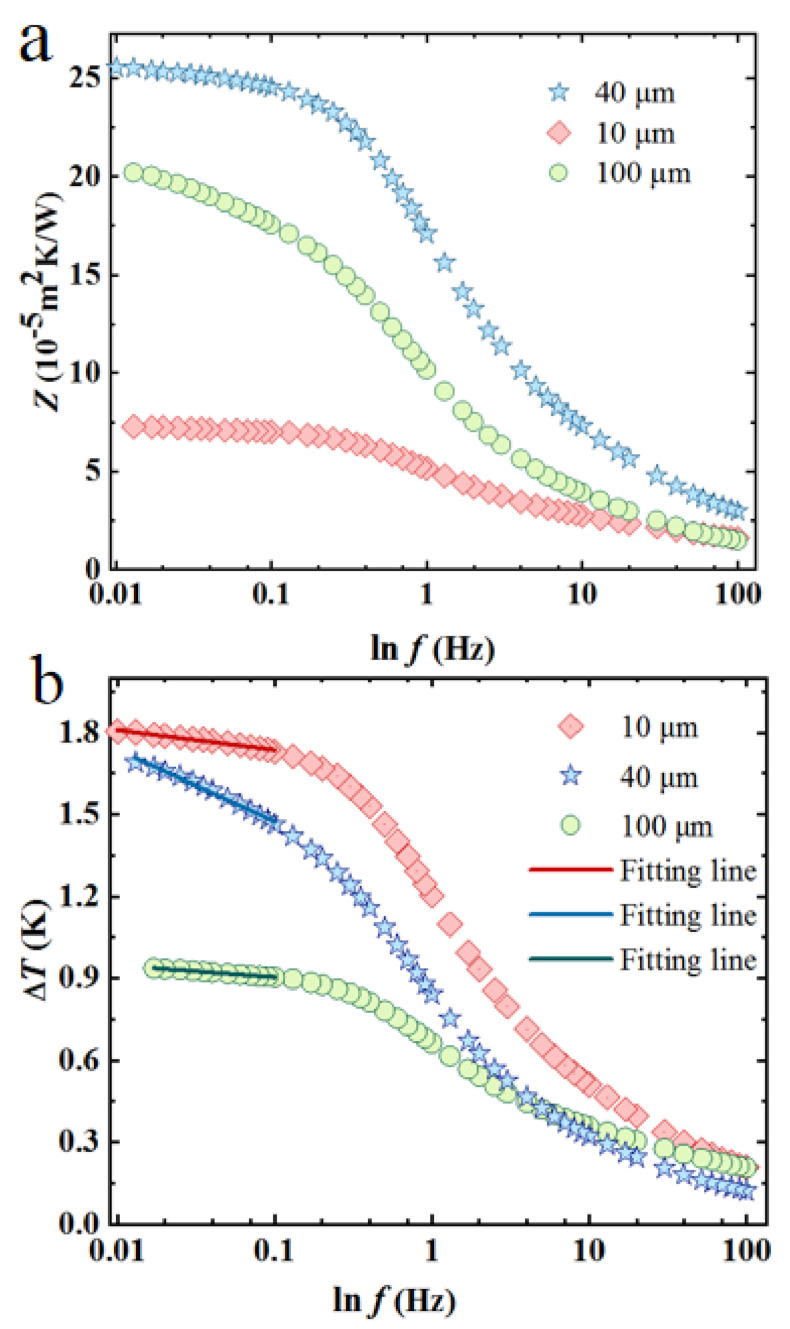
(**a**) The thermal impedance versus the logarithmic frequency. (**b**) The fitted results of temperature rise as a function of frequency.

**Table 1 sensors-21-07968-t001:** Comparison of the measured specimen thermal conductivities to reference values of [19] by the laser flash method.

Direction	$\mathrm{Measured}\;(Wm^{-1}K^{-1})$	Ref [19] $\left(Wm^{-1}K^{-1}\right)$	Remarks
x	1.63 ±0.06	1.67	
y	1.38 ±0.07	1.47	$49^{{}^{\circ}}$ tilt
		1.58	$31^{{}^{\circ}}$ tilt
z	1.32 ±0.09	1.35	

## Data Availability

Date availability on request.

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
