# Peer review of "Freestanding Flexible Sensor Based on 3ω Technique for Anisotropic Thermal Conductivity Measurement of Potassium Dihydrogen Phosphate Crystal"

_sensors, 2021, doi:10.3390/s21237968_

Round 1
Reviewer 1 Report
In this work, the authors fabricated a freestanding flexible sensor based on the 3ω method principle, and measured the thermal conductivities of potassium dihydrogen phosphate (KH2PO4, KDP) along the three principal axes. The work of depositing a thin film heater/temperature sensor on a sample is challenging, and the freestanding flexible sensor may contribute to the improvement of existing 3ω technique. The manuscript is well-written, while there are still several minor issues, as listed below, to be addressed before publication.
- The frequency region of Fig. 2(b) should be the same as Fig. 2(a). This is because the temperature rises at the lower frequency region are used to estimate thermal conductivity, and such a plot would be helpful for readers who want to figure out the results of the authors’ measurement. It also looks more concise.
- On page 4, line 6, it is said “The nickel patterns have 1.6 times larger thermal coefficient of resistance (TCR) than platinum during the temperature range of 0-250C, which renders higher sensitivity[11].” The author need to add the temperature range in which the independent detector works.
- 13 has gone through an errata, please also cite the erratum of the Ref. 13 in Eq. (1) for readers who are not familiar to the 3ω method. D. G. Cahill, Erratum: Thermal conductivity measurement from 30 to 750 K: The 3ω method, Rev. Sci. Instrum. 61, 802 (1990), Review of Scientific Instruments 73, 3701 (2002).
Reviewer 2 Report
This manuscript presents a sensor to measure thermal conductivities and results for PDP crystal. The approach and preliminary results are interesting but major issues need to be addressed prior to publication.
First, the results and discussion section is short and weak. The authors need to bolster this section.
Can this sensor be used to measure thermal conductivities of other materials? or is it just inherent to only PDP?
Why is there a dependence on strip width? Isn’t thermal conductivity just a material property?
The authors say under Section 3: “ The sensor serves as both a heater and a thermometer [12 ].” Actually, there is a better and more appropriate reference than ref. 12.
“ A built-in temperature sensor in an integrated microheater “, IEEE Sensors Journal, Vol. 16(14), pp. 5543-5547 (2016). This does not use any exotic material like carbon nanotubes as in ref. 12; in addition, it is a simple construction. So, the authors should add this reference to provide a more appropriate and accurate source of information.
Finally, the English is not good throughout the paper and the manuscript needs editorial help to improve the writing.
Round 2
Reviewer 2 Report
The revisions are satisfactory.